# With Great Ecosystem Services Comes Great Responsibility: Benefits Provided by Urban Vegetation in Brazilian Cities

**DOI:** 10.3390/plants14030392

**Published:** 2025-01-28

**Authors:** Helder Marcos Nunes Candido, Theodore A. Endreny, Fabrício Alvim Carvalho

**Affiliations:** 1Graduate Program in Biodiversity and Nature Conservation, Universidade Federal de Juiz de Fora, Juiz de Fora 36036-900, MG, Brazil; 2Department of Environmental Resources Engineering, College of Environmental Science and Forestry, State University of New York, Syracuse, NY 13210, USA; te@esf.edu; 3Departamento de Botânica, Instituto de Ciências Biológicas, Universidade Federal de Juiz de Fora, Juiz de Fora 36036-900, MG, Brazil; fabricio.alvim@gmail.com

**Keywords:** carbon sequestration, carbon stock, i-Tree Canopy, urban green spaces, vegetation benefits

## Abstract

Ecosystem services (ESs) are extremely important, specifically in urban areas. Urban forests, even representing a pivotal role in global sustainability, have been converted into different human-modified landscapes. This paper aims to analyze the ES provided by the urban areas of 25 cities of the Atlantic Forest in Brazil. We used i-Tree Canopy v.7.1 to classify the land use. We quantified the monetary benefits of the urban vegetation and used socioeconomic variables (i.e., total population, population density, Human Development Index (HDI), and Gross Domestic Product (GDP) per capita) to analyze if the ecosystem services or the land uses are associated with this. Our data reveal that together, the cities studied sequester a significant total of 235.3 kilotonnes of carbon and a substantial 864.82 kilotonnes of CO_2_ Equivalent (CO_2_ Equiv.) annually. Furthermore, together, they also store a total of 4861.19 kilotonnes of carbon and 17,824.32 kilotonnes of CO_2_ Equiv. We found out that the average monetary estimate of annual carbon sequestration was USD 3.57 million, while the average stored estimate was USD 73.76 million. Spearman’s correlogram showed a strong positive correlation between density and the percentage of impervious cover non-plantable no trees (IN) in urban areas (*p* < 0.001). IN was also positively correlated with HDI (*p* = 0.01), indicating that urban areas with higher HDI tend to have larger impervious areas. Our data suggest essential insights about the ecosystem services provided by urban areas and can serve as significant findings to drive policymakers’ attention to whether they want to provide more ecosystem services in cities.

## 1. Introduction

In a changing world, conservation efforts are mandatory to pursue a more sustainable future, since trees can provide several valuable benefits to people who live in urban spaces. In this context, ecosystem services are defined by the contribution that ecosystems give to humans, whether directly or indirectly [1]. Ecosystem services can be of a wide range, starting with basic life support systems for humankind (climate and air quality regulation), material goods (food, energy, medicines), and spiritual, cultural, and recreational uses [2]. In urban landscapes, trees can help maintain hydrological functions, such as precipitation interception, water infiltration, and evapotranspiration, which will reflect on soil moisture, water quality, and erosion mitigation [3].

Urban vegetation and their ecosystem services are frequently undervalued in urban planning and policymaking, even though research has demonstrated their critical role in enhancing public health and well-being. For instance, Wolf et al. [4] estimated that urban forests remove 17.4 million tons of air pollutants annually in the United States, significantly reducing the incidence of respiratory and cardiovascular illnesses. Similarly, Gascon et al. [5] observed that living near urban green spaces is linked to a 20% decrease in self-reported stress levels and notable improvements in mental health. Additionally, urban trees play an integral role in urban heat island mitigation, offering shade and cooling effects that can reduce energy costs for cooling systems [6]. These benefits underline the need to better integrate urban forest management into broader sustainability strategies.

Even though they have a pivotal role in global sustainability, due to uncontrolled economic development, natural environments have been converted to different human-modified landscapes due to multiple land use purposes, such as increased demand for agricultural land, livestock grazing, mining, energy, and other destination [7,8,9,10]. Environmental degradation exacerbates climate change, as deforestation and habitat destruction reduce the Earth’s capacity to sequester carbon, leading to increased greenhouse gas emissions [11]. Addressing these challenges requires a comprehensive approach that balances economic development with environmental conservation, ensuring that natural ecosystems are conserved for their intrinsic value and the essential services they provide to humanity.

Therefore, managing biodiversity is one of the main priorities of today, and investigating environmental data is increasingly important for future conservation and preservation purposes. In this way, Geography Information Systems (GISs) work as a strong ally in the area of natural sciences to contribute to the survey and analysis of environmental data with high applicability to promote public policies. As mentioned by Riley & Gardiner [12], in the past few years, a large number of tools have been developed to quantify ecosystem services and their value. One of the most popular and used for urban forests worldwide is the i-Tree platform (https://www.itreetools.org/), a free suite of software programs backed by peer-reviewed research that allows obtaining estimates of the ecosystem services and monetary value of UPFs based on a variety of data collection techniques. For example, the i-Tree Canopy [13] program uses a methodology to produce a statistically valid estimate of land cover use through aerial images available in Google Maps and provides some important outputs like the quantity of pollutants removed (and their equivalent monetary value based on USA medical services), the quantity of atmospheric carbon (CO_2_ Equivalent) removed and the total carbon stock in the vegetation, as well those monetary values. Then, GIS tools represent unique applications in surveying urban forests, whose ecosystem contributions can be estimated in a more rapid and low-cost way [14].

The implementation of the Paris Climate Agreement in 2015 accelerated the demand for carbon credits, with their market value steadily increasing in recent years. This trend highlights the potential of urban vegetation to generate economic benefits while addressing global climate challenges. The economic valuation of biodiversity is increasingly recognized as a crucial component for effective conservation strategies, as highlighted by Agliardi et al. [15], who emphasize that biodiversity contributes significantly to industries such as agriculture, pharmaceuticals, and tourism. Currently, several fintech, traditional banks, and investment brokers have negotiated carbon credits in the voluntary market since the number of companies interested in environmental policies has grown in recent years. Furthermore, the carbon market is extremely heated after the implementation of the Paris Climate Agreement in 2015 and tends to increase even more. Then, carbon markets are gaining relevance; for example, the unit of Carbon Emissions Future stocks (CFI2Z1), traded on the London Stock Exchange, was traded at approximately EUR 24.00 at the beginning of 2020 and in April 2021 was traded at EUR 94.31 [16].

The rapid urbanization in the Brazilian Atlantic Forest region highlights the need to address threats to urban vegetation and its ecosystem services since land use might impact the biome phylogenetic diversity and plants’ habitat suitability in a climate change scenario [17,18]. Therefore, considering the incredible biodiversity that Brazilian forests have, even when located in cities, measures for their conservation and protection are crucial for a truly sustainable future. The need for correct environmental management is even more urgent given the various ecosystem services promoted by vegetation, which can greatly benefit human well-being and have significant economic importance. Considering that carbon sequestration and storage are one of the primary ecosystem services today, as they reduce the concentration of greenhouse gases in the atmosphere, more studies should be carried out to promote public policies aimed at mitigating global climate change, which tend to be disastrous for several species, including human beings. Among the mechanisms that support environmental studies and decision-making, GIS is accessible, requires low investments, and, even so, generates accurate products for better urban environmental management. Among GIS tools, i-Tree Canopy stands out, performing, through satellite images, the estimation of ecosystem services promoted by the vegetation of cities, especially concerning the removal of greenhouse gases, and is widely used [19,20,21]. In this sense, with the global growth trend of carbon markets, such measurements are essential, mainly due to the possible contribution of urban vegetation in the carbon emissions trading market.

The goal of this manuscript is to estimate the ecosystem services promoted by the vegetation of the urban area of 25 Brazilian cities of the Atlantic Forest. Therefore, to achieve this goal, the following specific objectives are proposed: (1) mensurate annual ecosystem services of pollutant removal, carbon sequestration, and carbon stocks by vegetation; (2) evaluate the economic benefits of these ecosystem services via carbon markets; and (3) assess if socioeconomic variables have any relationship with land cover classes and pollutants and carbon sequestration.

## 2. Results

### 2.1. Land Cover

Our results show that the dominant land cover class was “Impervious cover non-plantable no trees” (IN), with 80% of the cities studied having IN as the dominant land cover in the urban area. The average impervious area in urban areas was 47.02% (Figure 1). IN values varied from 22.37% in Camaçari, in the state of Bahia, to 76.21% in Diadema, a city in the metropolitan region of São Paulo, the most populated Brazilian city (Table 1).

The cities of Campos dos Goytacazes, in Rio de Janeiro state, Simões Filho, in Bahia state, and Macaé, in Rio de Janeiro state, presented “Short Vegetation cover partially plantable no the trees” (SVP) as the dominant cover class in their central regions, with 34.44%, 30.27%, and 35.52%, respectively (Figure 2).

Blumenau, located in the state of Santa Catarina, is the only city in which the “Tree Evergreen over Pervious” cover class dominates, accounting for 45.71% of the urban areas. The average presented for the “Tree Evergreen over Pervious” cover class was 18.20% (Figure 3). Table 1 illustrates the other land cover classes and their percentage variation through the cities.

### 2.2. Pollutants Removal

Our data reveal that about 5739.66 tons of the six air pollutants are sequestered annually by vegetation in the cities studied. The city of Taboão da Serra (SP) presented the lowest values for the sequestration of pollutants in this study. On the other hand, surprisingly, Porto Alegre (RS), a state capital, presented the highest amounts for pollutant sequestration annually (Table 2).

Carbon monoxide (CO) sequestered annually varied from 0.35 tons in Taboão da Serra (SP) to 15.67 tons in Porto Alegre. The average amount of CO sequestered annually among the cities studied is 4.08 tons.

Concerning the results found for nitrogen dioxide (NO_2_), the amount sequestered annually ranged from 1.37 to 60.48 tons. The overall average was 15.75 tons. Regarding the amount of ozone (O_3_) sequestered annually, our data range from 13.14 to 580.21 tons, averaging 151.20 tons. The annual sulfur dioxide (SO_2_) sequestration varied between 0.74 and 32.86 tons. The average presented was 8.56 tons.

The values of particulate matter removed less than 2.5 microns (PM_2.5_) and ranged from 0.69 to 30.67 tons annually. With respect to particulate matter greater than 2.5 microns and less than 10 microns (PM_10_) removed annually, we had a range between 3.65 and 161.1 tons.

### 2.3. Carbon Sequestration

Together, the cities studied sequester a significant total of 235.3 kilotonnes of carbon and a substantial 864.82 kilotonnes of CO_2_ Equivalent (CO_2_ Equiv.) annually. Furthermore, together they also store, through their vegetation, a total of 4861.19 kilotonnes of carbon and 17,824.32 kilotonnes of CO_2_ Equiv. Once again, the city of Taboão da Serra presented the lowest values in all parameters sampled. Porto Alegre recorded the highest values for the carbon and CO_2_ Equiv. parameters. The average of carbon sequestered annually was 9.41 kilotonnes per city and 34.59 kilotonnes of CO_2_ Equivalent. For each city, the average carbon stocked by the vegetation was 194.44 kilotonnes, while the average CO_2_ Equiv. value stored was 712.97. All the parameters analyzed, for all cities, can be found in Table 3.

### 2.4. Monetary Measurement

When it comes to the monetary estimate of the ecosystem services promoted by the 25 cities studied, together, they are estimated to contain a total of USD 1.8 billion in carbon credits stocked, sequestering a total of more than USD 89 million in carbon credits annually (Table 4).

Again, our data points to the city of Taboão da Serra (USD 0.3 million sequestered annually; USD 6.4 million stocked) as having the lowest potential values to be applied in carbon credits and the city of Porto Alegre (13.7 million sequestered annually; 283 million stocked) as having the highest values, mirroring previous results (Figure 4 and Figure 5). The average monetary estimate of annual carbon sequestration was 3.57 million, while the average stored estimate was USD 73.76 million.

### 2.5. Variables Correlations

After conducting Spearman’s correlation analysis, the correlogram showed a strong positive correlation between density and the percentage of impervious areas (IN) in urban areas (*p* < 0.001) (Figure 6 and Figure 7). IN was also positively correlated with HDI (*p* = 0.01), indicating that urban areas with higher HDI tend to have larger impervious areas. The cover classes and socioeconomical variables that are highly correlated are presented in Figure 8.

Our results also show a positive correlation between population and “short vegetation non-plantable no trees” (SVN) (*p* < 0.01), indicating that the increase in population influences the amount of this class. A negative relationship is also shown between “short vegetation cover partially plantable no trees” (SVP) and HDI (*p* < 0.01). SVP shows a negative correlation with HDI (*p* < 0.05).

Our findings indicate a negative correlation between the amount of “soil cover partially plantable no trees” (SP) in urban areas and both population density and total population (*p* < 0.01). The results suggest that as population and density increase, there are fewer areas available for planting new trees.

## 3. Discussion

### 3.1. Pollutant Sequestration and the Role of Urban Vegetation in Air Quality Improvement

In this study, we analyzed, in urban areas, the cover classes’ relationships with socioeconomic variables and the contributions in pollutant removal by the urban vegetation of several cities. Our findings demonstrate that, in most of the cities, the urban areas provide valuable ecosystem services to the human population and biodiversity. Therefore, our study underscores the importance of urban vegetation in promoting environmental sustainability and improving the quality of life in urban areas. We found that more than 235,000 tons of carbon is sequestered annually, and more than 4 million tons are stored in the vegetation of the studied cities. Furthermore, regarding CO_2_ Equiv., about 864,000 tons of this pollutant are sequestered annually, and more than 17 million tons are stored in trees. The remarkable results show that vegetation can substantially impact climate change mitigation and improve air quality [22,23].

Our data indicate that the amount of CO_2_ sequestered by vegetation in urban areas can vary greatly between cities, with Porto Alegre and Blumenau standing out as particularly effective areas for carbon sequestration, which can be due to the fact that these two cities have the largest quantity of vegetation in their urban areas. The variation in CO_2_ sequestration rates between cities may be due to a variety of factors, including differences in vegetation coverage, types of vegetation, and environmental conditions. Therefore, this information underscores the importance of protecting and promoting urban green spaces for their role in carbon sequestration and mitigating the impacts of climate change, but we also highlight that proper planning for these urban areas is mandatory for the vegetation to achieve its potential [24]. The variation in the amount of CO_2_ sequestered by urban vegetation in different cities highlights the importance of local factors in determining the effectiveness of urban green space for pollutant sequestration [24]. Factors such as the type, structure, and density of vegetation, the amount of impervious surfaces in the city, the level of vehicular traffic, and spatial pattern optimization of urban green space can all have significant impacts on the ability of urban vegetation to absorb pollutants like CO_2_ [25].

Our findings regarding the amount of carbon and CO_2_ sequestered annually by urban vegetation in Brazilian cities are significant for understanding the role that urban green spaces have in cities. By investing in urban green space and promoting sustainable urban development, cities can harness the power of nature to improve the welfare of their residents, while also contributing to global efforts to mitigate the impacts of climate change [26]. The variation in pollutant sequestration between cities is likely due to a combination of factors, such as differences in land cover, vegetation type and density, and local emissions sources. Taboão da Serra has a higher proportion of impervious surfaces, such as roads and buildings, compared to other cities in the study (70.15%) which could limit the amount of vegetation cover and, therefore, the capacity for pollutant sequestration. In contrast, Porto Alegre may have more vegetation cover and probably a more diverse range of vegetation types that are effective at sequestering pollutants. Additionally, the city has implemented policies and programs, such as a Master Plan for Urban Forestry, to promote the planting and maintenance of urban trees and other vegetation [27].

The results of the study show that urban vegetation plays an important role in sequestering pollutants. NO_2_ is a toxic component of urban air that can cause several respiratory problems [28]. It is mainly produced by fossil fuel combustion, particularly in transportation, electricity generation, and industrial and residential activities [29,30]. Therefore, cities with higher levels of vehicular traffic or industrial activity may have higher levels of NO_2_ emissions, which could affect the capacity of vegetation to sequester this pollutant. In contrast, cities with lower levels of traffic and industrial activity may have fewer NO_2_ emissions, which could increase the effectiveness of vegetation in sequestering this pollutant. O_3_ is a pollutant that is formed in the atmosphere through the reaction of nitrogen oxides (NOx) and volatile organic compounds (VOCs) in the presence of sunlight [31]. Therefore, the capacity of vegetation to sequester O_3_ may be influenced by factors such as the abundance and types of vegetation, the intensity and duration of sunlight, and the levels of NOx and VOCs in the atmosphere, which have increased in urban areas due to vehicles [32].

The removal of particulate matter (PM) by vegetation is a critical ecosystem service provided in urban areas [33]. The results of this study suggest that vegetation in urban areas of the cities studied can effectively remove both PM_2.5_ and PM_10_, which are known to have adverse effects on human health and the environment, specifically because the presence of particulate matter (PM) in the air presents a greater risk to human health compared to ground-level ozone and other commonly found air pollutants, such as carbon monoxide [34]. The removal of PM_2.5_ and PM_10_ by vegetation can occur through a variety of mechanisms, including dry deposition, interception, and absorption [35,36]. Vegetation can capture and retain particles through the physical structure of leaves and branches, as well as through the uptake of particles by stomata and the absorption of particles through the leaf surface [37,38]. The range of values found for PM_2.5_ and PM_10_ removal in this study reflects differences in the abundance and quality of vegetation in the urban areas studied, as well as differences in local sources of PM emissions. Cities with higher levels of vehicular traffic or industrial activity may have higher levels of PM emissions, which could affect the capacity of vegetation to remove these pollutants [39].

### 3.2. Economic Potential of Urban Trees Through Carbon Credits

Another important finding is related to the great economic potential that urban trees can mean regarding carbon credits. Even though our study was conducted only in urban areas, which may be tiny in smaller cities, it points to great economic potential, which can serve as data for decision-makers to conserve and prioritize investment in green infrastructure and to inform urban planning and policy. The average values of the estimated annual sequestration exceed 3 million dollars when considering only the urban area of the studied municipalities. Furthermore, the average estimated values for the carbon that is stored in the vegetation are more than 73 million dollars. The finding that urban trees have great economic potential in terms of carbon credits is expressive as it provides an incentive for decision-makers to invest in green infrastructure and prioritize urban tree conservation.

It is interesting to note that even though the average results for annual sequestration and carbon stock were lower compared to the results found in an urban study in the city of Naples, Italy [40], our average study area (urban area of each municipality) is much smaller than the 1200 Km^2^ studied by the Italians, who measured about USD 24 million in annual sequestration and USD 530 million for the carbon stored in trees. It is worth mentioning that, besides having a different total area, the economic measurement of the Italian study and the present research was also different.

### 3.3. Land Cover Dynamics

The results of our analysis show that the dominant land cover class in the urban areas of the Brazilian cities studied is “Impervious cover non-plantable no trees” (IN), with 80% of the cities having IN as the dominant land cover. The findings are consistent with research showing that urbanization is often associated with increases in impervious surfaces, such as roads, parking lots, and buildings, which can negatively impact ecosystem services and urban biodiversity [41,42]. It is particularly concerning that the values of IN varied widely across the cities studied, ranging from 22.37% in Camaçari to 76.21% in Diadema. The results suggest that some cities may be more vulnerable to the negative impacts of impervious surfaces than others, and that targeted efforts may be needed to address this issue. One potential solution for reducing the negative impacts of impervious surfaces in urban areas is to increase the amount of urban green space, including trees and other vegetation.

Some cities, like Campos dos Goytacazes, Macaé, and Simões Filho, presented SVP as the most dominant land cover in the urban areas. Our findings are particularly interesting because they demonstrate that above mentioned cities have large areas that can potentially serve for planting arboreal vegetation, which tend to promote a greater amount of ecosystem services related to pollutant sequestration and carbon stock when compared to short vegetation and can have significant benefits for public health, air quality, and climate change mitigation. It is worth noting, however, that planting and maintaining urban trees can be challenging, particularly in areas with high levels of impervious surfaces or other competing land uses [43,44]. As such, any efforts to increase urban tree cover in these cities, preferably with native and non-invasive species, should be accompanied by careful planning and management, as well as community engagement and education [45]. Our finding states that Blumenau is the only city where the dominant cover class is “Tree Evergreen over Pervious”, which is particularly noteworthy because it highlights the potential for other cities in Brazil to increase their urban tree cover and promote sustainable urban development.

### 3.4. Urban Development and Ecosystem Services

The positive correlation between density and IN could be explained by the fact that cities with higher density tend to have more buildings and less available space for vegetation, resulting in larger impervious areas [46,47], which could negatively impact the provision of ecosystem services, such as air quality regulation and carbon sequestration, as impervious surfaces prevent water infiltration into the soil and reduce the ability of trees and other vegetation to absorb pollutants and store carbon. The positive correlation between IN and HDI could be attributed to the fact that cities with higher HDI tend to be larger [48], which can result in a greater proportion of impervious surfaces. However, the positive correlation may also indicate a potential trade-off between urban development and the provision of ecosystem services, as more developed areas tend to have a higher demand for infrastructure and may prioritize economic development over environmental considerations.

### 3.5. Challenges of Urban Greening and Future Directions

Overall, the results suggest that urban planning and management strategies should consider the potential trade-offs between urban development and the provision of ecosystem services, and aim to balance economic, social, and environmental objectives. Strategies such as green infrastructure planning, green roofs, and urban forests can help mitigate the negative impacts of impervious surfaces and increase ecosystem services in urban areas.

The findings highlight the importance of considering urban population dynamics in the context of carbon management strategies. Our findings highlight the challenges of urban greening and the importance of considering the availability of suitable planting spaces in urban planning. As cities continue to grow and populations increase, there is a greater need for green spaces to enhance the quality of life of urban residents and mitigate the negative impacts of urbanization. However, the limited availability of soil for planting new trees can make it challenging to achieve this goal. Therefore, policymakers and urban planners need to consider innovative approaches to create new green spaces.

## 4. Materials and Methods

### 4.1. Study Area

For data collection, we first set a threshold by selecting all 226 cities in the Atlantic Forest with an estimated population of over 100,000 inhabitants in 2020, according to the IBGE [49], because such regions typically experience significant urban anthropogenic pressures and socioeconomic factors, with major urban expansion processes strongly linked to the decline of urban forests and their ecosystem services [50,51,52,53]. Subsequently, we divided the cities into five groups based on population (Table 5). Then, we randomly selected five cities from each group, resulting in 25 studied cities (Figure 8).

**Figure 8 plants-14-00392-f008:**
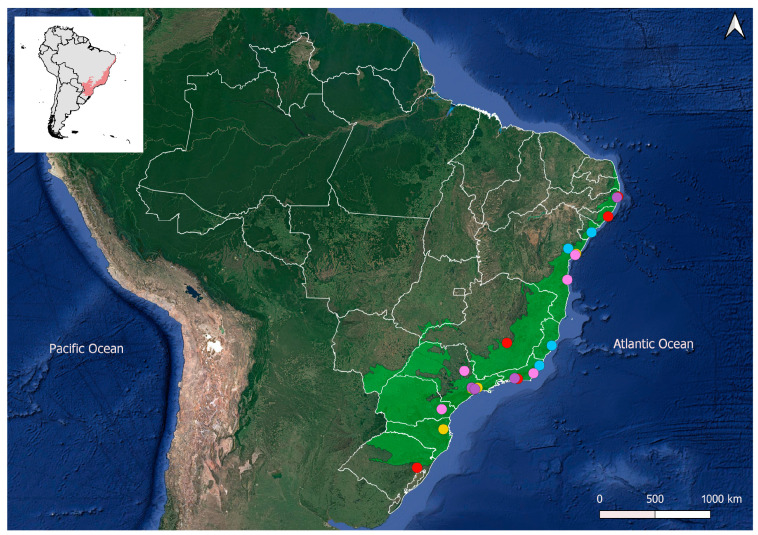
Distribution of the studied cities. Red dots = cities with more than 1 million inhabitants; purple dots = cities with 700,000–900,000 inhabitants; blue dots = cities with 500,000–700,000 inhabitants; yellow dots = cities with 300,000–500,000 inhabitants; pink dots = cities with 100,000–300,000 inhabitants; green shadow = Atlantic Forest.

### 4.2. Tree Cover Assessment

To assess tree canopy coverage (area and percentage) and classify the land cover in urban areas, we utilized the land classification algorithm tool [13], which is part of the i-Tree tools suite and is freely distributed by the United States Department of Agriculture. The i-Tree Canopy is a web-based tool that employs the random sampling method, which facilitates efficient data acquisition. Moreover, the vegetation data obtained through i-Tree tools using satellite imagery exhibit a strong correlation (R^2^ = 0.9) with results obtained via advanced and expensive tools, such as airborne Light Detection and Ranging (LiDAR), as reported by [54].

Since our goal is to evaluate the ecosystem services promoted by the urban area of the studied municipalities, in this study, we used the classification of urbanized areas in Brazilian cities, made available by [49]. Therefore, the boundaries of the urban areas were cut using the “clip” tool in ArcGIS v 10.6 [55]. We input the ESRI shapefile into i-Tree Canopy. As the program uses an algorithm based on metrics from the USA, for this work, we selected only the American states that have a Köppen climate classification [56] similar to that found in the Atlantic Forest, to avoid inaccuracies in the results. The American states selected were Alabama, Arkansas, Florida, Georgia, Louisiana, North Carolina, and Tennessee. Only metrics for urban areas available in the software were considered. This was carried out for all 25 cities.

After that, we performed a photo-interpretation of the random points generated by the program. We performed the photo-interpretation using high-resolution aerial imagery recorded in 2022 and 2023 from the National Center for Space Studies/Airbus satellites made available by Google Earth. We generated random points until we found a standard error ≤ 2.5%, which measures uncertainty in land cover within the software, for each land cover category. The cover classes used in this study can be found in Table 6.

### 4.3. Ecosystem Services

Based on the estimated vegetation cover area provided by i-Tree Canopy, we assessed the following ecosystem services: total carbon sequestered annually in trees; total carbon stored in trees; and removal of 6 air pollutants, which are Carbon Monoxide removed annually; Nitrogen Dioxide removed annually; Ozone removed annually; Sulfur Dioxide removed annually; Particulate Matter less than 2.5 microns removed annually and Particulate Matter greater than 2.5 microns and less than 10 microns removed annually. Carbon and carbon equivalent estimations are presented in kilotonnes (kt), while the other parameters are presented in tons (t).

### 4.4. Monetary Valuation of Ecosystem Services

With the analysis of satellite images, the program measures certain ecosystem services facilitated by urban vegetation while estimating the monetary value of ecosystem services and their significance to the community [57]. The developers of the software used various metrics related to the costs of pollution impacts on human health, such as reduced productivity, hospital admissions, and mortality, to achieve one of the goals of this study—the monetary valuation of air pollutant removal, including CO_2_ Equivalent [58]. Valuing ecosystem services requires consideration of various factors, including healthcare and labor legislation. In Brazil, we have a universal healthcare system called the Unified Health System (SUS) that differs from the American healthcare system, and we also have unique labor legislation. Therefore, to estimate ecosystem services economically, we used an alternative approach of equivalent carbon dioxide (CO_2_ Equiv.) values based on the carbon credit market, as proposed by Costemalle et al. [21], which fits better in the Brazilian reality. Then, we utilized values from the ’CFI2Z1-Future Carbon Credit’ market. For this, we consider the value of CFI2Z1 recorded on 18 April 2023 (EUR 94.31). The values were later converted into dollars following the exchange rate of the same day (1 euro = 1.0971 dollars).

### 4.5. Socioeconomic and Population Data

Socioeconomic and population factors, such as wealth, population density, and Human Development Index (HDI), are crucial for understanding the distribution and functionality of green spaces in urban areas [59,60]. Additionally, urban vegetation landscape structure is shaped by population dynamics, socioeconomic conditions, climate, and topographic features, with economically unequal and developing regions often experiencing more fragmented vegetation patterns [61]. Therefore, socioeconomic and population factors, such as wealth and population density, may correlate with the greater or lesser provision of green areas in cities [62,63,64]. For instance, cities with higher HDI or GDP per capita may have more resources to invest in urban greening, while higher population densities often correlate with increased impervious surfaces, potentially limiting vegetation growth and associated ecosystem services [47]. To explore these relationships, we collected data on total population, population density (Km^2^), HDI, and GDP per capita (BRL) from IBGE’s ‘Cities’ portal [65]. These variables were selected to assess their potential correlations with land cover classes and pollutant removal, thereby providing insights into how urban planning and socioeconomic conditions shape the provision of ecosystem services. HDI and GDP per capita were selected due to the fact that a city with high values for these variables tends to have greater capacity and budget available for investment in green infrastructure. Population density was selected because cities with a higher population density tend to have a larger impermeable area and greater impacts on urban vegetation [66].

### 4.6. Data Analysis

We conducted a correlation analysis to investigate the relationships between several factors, including the HDI, total population, population density, GDP, and various pollution parameters estimated using the iTree Canopy tool. To analyze the correlations, we used the Spearman algorithm. We visualized the results through the ggcorrplot package [67] in R software [68].Then, the trends were analyzed using scatterplots, also in R [68], using the Kendall method.

## 5. Conclusions

Our results show that the amount of carbon, CO_2_ Equivalent, and other pollutants sequestered and stored by urban vegetation can vary greatly between cities and is influenced by factors such as the amount of vegetation present, total population, population density, and land use patterns. The results of the average of pollutants sequestered annually and stored among the cities studied provide a useful benchmark for assessing the effectiveness of urban green space in different cities. Cities with lower amounts of pollutants sequestered may want to consider strategies for increasing the amount of urban vegetation, while cities with higher amounts may want to build on their existing green space to further promote the ecosystem services provided by trees and other vegetation. The fact that the city of Taboão da Serra consistently presented the lowest values for all parameters sampled suggests that there may be opportunities for this city to increase the amount of vegetation present in its urban areas, potentially leading to greater ecosystem service provision. On the other hand, the city of Porto Alegre, which had the highest values for the carbon and CO_2_ Equiv. parameters, may be seen as an example of how urban vegetation can provide significant benefits in terms of carbon storage and pollutant sequestration. By highlighting the variation in ecosystem services provided by urban vegetation across different cities, the study provides important insights for policymakers and urban planners seeking to promote urban greening initiatives.

## Figures and Tables

**Figure 1 plants-14-00392-f001:**
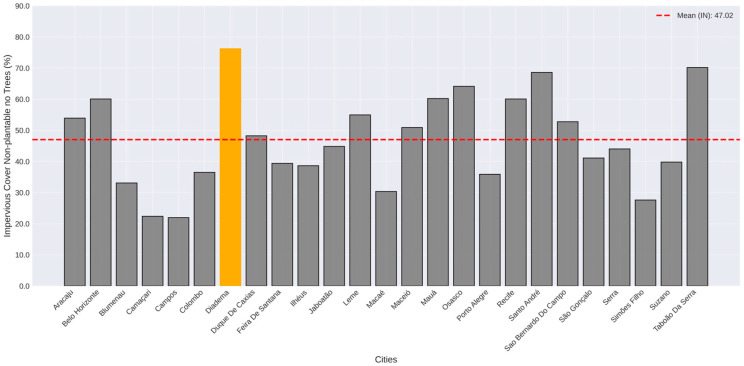
Impervious cover non-plantable no trees (IN) percentages in the city center of the studied cities. Dotted red line represents the average value. Yellow bar represents the maximum value.

**Figure 2 plants-14-00392-f002:**
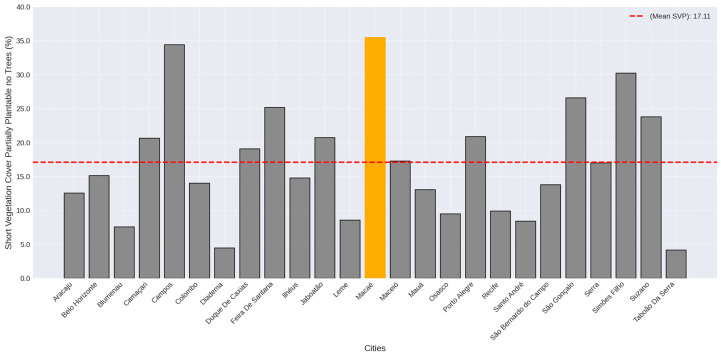
Short Vegetation cover partially plantable no trees (SVP) percentages in the city center of the studied cities. Dotted red line represents the average value. Yellow bar represents the maximum value.

**Figure 3 plants-14-00392-f003:**
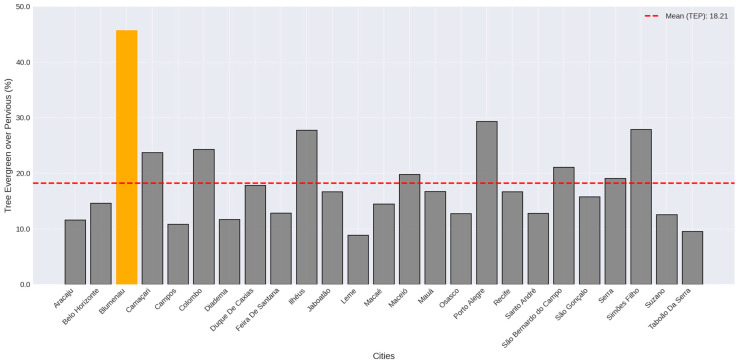
Tree Evergreen over Pervious (TEP) percentages in the city center of the studied cities. Dotted red line represents the average value. Yellow bar represents the maximum value.

**Figure 4 plants-14-00392-f004:**
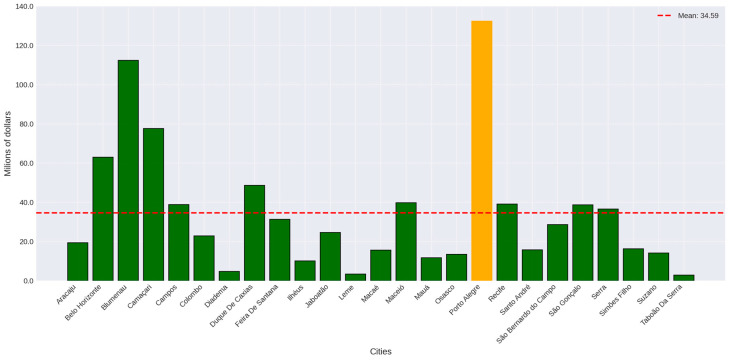
Monetary estimate of the annual CO_2_ sequestration realized by the vegetation of the city center of the studied cities. Dotted red line represents the average value. Yellow bar represents the maximum value.

**Figure 5 plants-14-00392-f005:**
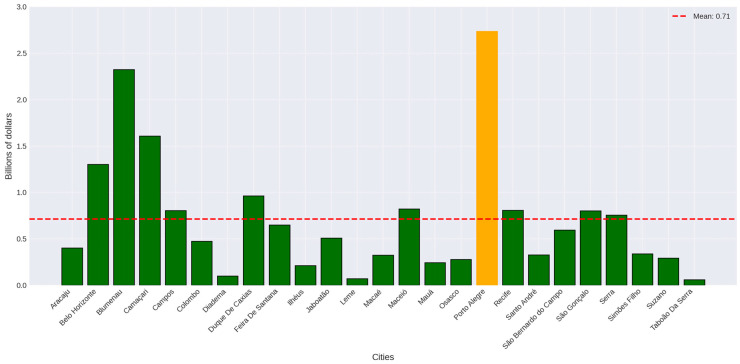
Monetary estimate of the total CO_2_ stocked in the vegetation of the city center of the studied cities. Dotted red line represents the average value. Yellow bar represents the maximum value.

**Figure 6 plants-14-00392-f006:**
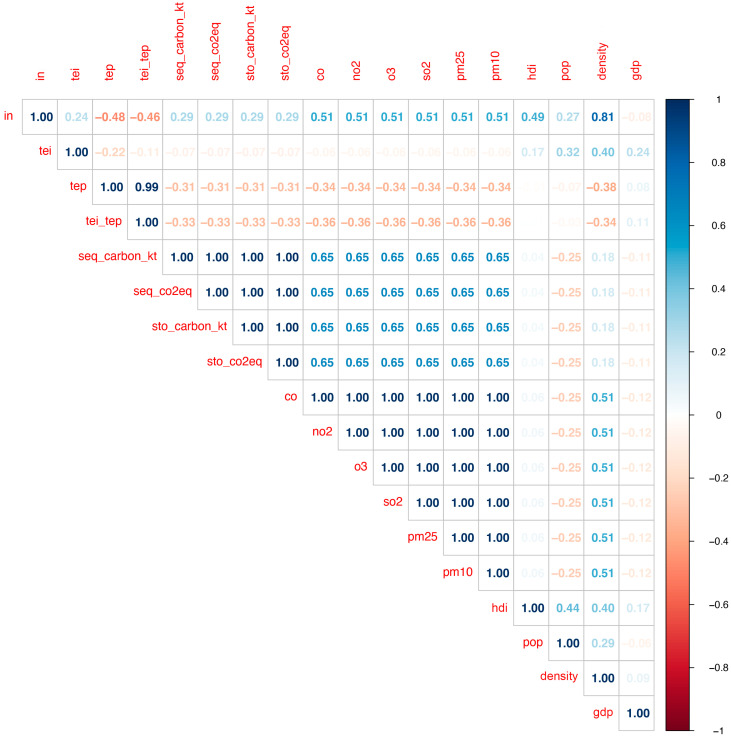
Correlation analysis chart between land cover classes, pollutants, and socioeconomic variables. In = Impervious cover non-plantable no trees; tei = Tree evergreen over impervious; tep = Tree evergreen over pervious; tei_tep = Tree evergreen over impervious + tree evergreen over pervious; seq_carbon_kt = Carbon sequestered annually (Kt); seq_co2eq = Carbon equivalent sequestered annually; sto_carbon_kt = Carbon stocked (Kt); sto_co2eq = Carbon equivalent stocked; co = Carbon monoxide; no2 = Nitrogen Dioxide; o3 = Ozone; so2 = Sulfur Dioxide; pm25 = Particular matter smaller than 2.5 microns; pm10 = Particular matter between 2.5 and 10 microns; hdi = Human Development Index; pop = Total population; density = Population density; gdp = Gross domestic product per capita.

**Figure 7 plants-14-00392-f007:**
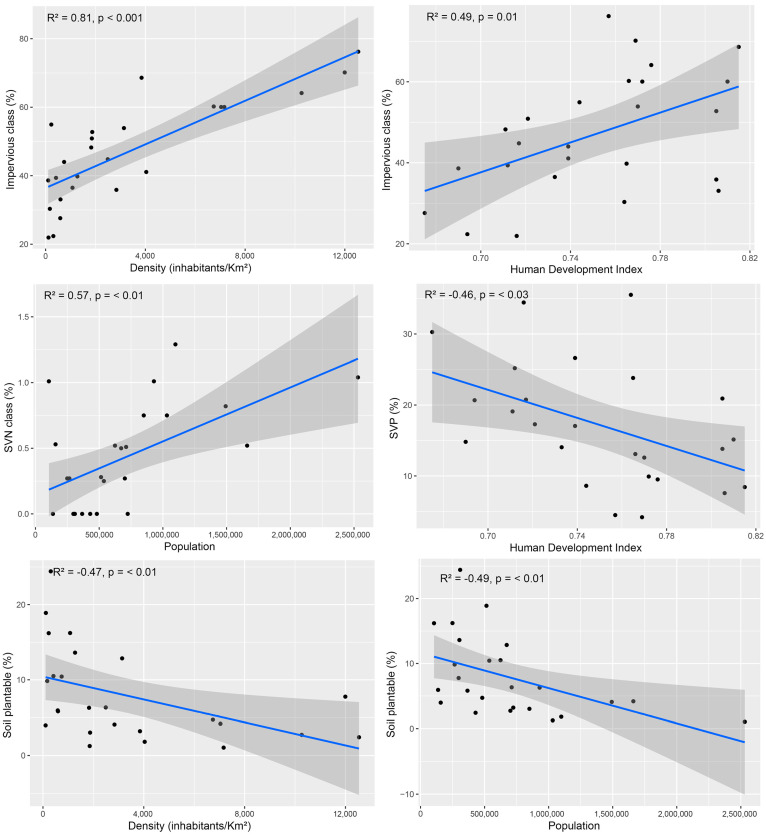
Simple linear regression plots with the relationships between land cover classes and socioeconomic variables. SVN = Short vegetation cover non-plantable no trees; SVP = Short vegetation cover partially plantable no trees.

**Table 1 plants-14-00392-t001:** Cover classes (%) found in the studied cities. IN = Impervious cover non-plantable no trees; IP = Impervious cover partially plantable no trees; SN = Soil cover non-plantable no trees; SP = Soil cover partially plantable no trees; SVN = Short Vegetation cover non-plantable no trees; SVP = Short Vegetation cover partially plantable no trees; TEI = Tree Evergreen over Impervious; TEP = Tree Evergreen over Pervious; W = Water, and OTH = Other.

City	IN	IP	OTH	SN	SP	SVN	SVP	TEI	TEP	W	TEI + TEP
Aracaju (SE)	53.9	0.25	1.26	3.53	12.85	0.5	12.59	0.76	11.59	2.77	12.35
Belo Horizonte (MG)	60.05	4.44	2.87	0	1.04	1.04	15.14	0	14.62	0.78	14.62
Blumenau (SC)	33.08	1.01	3.28	1.77	5.81	0	7.58	0.51	45.71	1.26	46.22
Camaçari (BA)	22.37	1.02	4.75	1.36	24.42	0	20.68	1.02	23.73	0.68	24.75
Campos dos Goytacazes (RJ)	21.94	1.11	3.33	4.17	18.89	0.28	34.44	1.94	10.83	3.06	12.77
Colombo (PR)	36.49	2.7	4.59	1.08	16.22	0.27	14.05	0	24.32	0.27	24.32
Diadema (SP)	76.21	1.72	2.07	0	2.41	0	4.48	1.03	11.72	0.34	12.75
Duque de Caxias (RJ)	48.24	1.01	2.51	1.51	6.28	1.01	19.1	1.51	17.84	1.01	19.35
Feira de Santana (BA)	39.37	2.1	4.2	3.41	10.5	0.52	25.2	1.31	12.86	0.52	14.17
Ilhéus (BA)	38.62	2.38	5.56	0.53	3.97	0.53	14.81	0	27.78	5.82	27.78
Jaboatão dos Guararapes (PE)	44.81	2.28	2.78	2.53	6.33	0.51	20.76	2.03	16.71	1.27	18.74
Leme (SP)	54.94	1.52	7.85	0.51	16.2	1.01	8.61	0	8.86	0.51	8.86
Macaé (RJ)	30.33	1.91	4.92	0.82	9.84	0.27	35.52	0.27	14.48	1.64	14.75
Maceió (AL)	50.88	2.76	4.26	1	1.25	0.75	17.29	1	19.8	1	20.8
Mauá (SP)	60.21	1.57	0.79	2.36	4.71	0	13.09	0.26	16.75	0.26	17.01
Osasco (SP)	64.13	5.16	1.9	0.27	2.72	0.27	9.51	2.99	12.77	0.27	15.76
Porto Alegre (RS)	35.87	3.53	1.63	0.54	4.08	0.82	20.92	2.45	29.35	0.82	31.8
Recife (PE)	60.05	2.35	0.78	0.78	4.18	0.52	9.92	3.13	16.71	1.57	19.84
Santo André (SP)	68.6	2.91	0.29	1.16	3.2	0	8.43	2.03	12.79	0.58	14.82
São Bernardo do Campo (SP)	52.76	0.75	2.51	2.01	3.02	0.75	13.82	0.25	21.11	3.02	21.36
São Gonçalo (RJ)	41.09	6.2	4.65	1.29	1.81	1.29	26.61	1.29	15.76	0	17.05
Serra (ES)	44.02	0.25	3.56	3.31	10.43	0.25	17.05	0.76	19.08	1.27	19.84
Simões Filho (BA)	27.6	0.89	6.53	0	5.93	0	30.27	0	27.89	0.89	27.89
Suzano (SP)	39.79	1.05	4.97	2.62	13.61	0	23.82	0.52	12.57	1.05	13.09
Taboão da Serra (SP)	70.15	3.58	2.69	0	7.76	0	4.18	1.79	9.55	0.3	11.34
**Average**	**47.02**	**2.17**	**3.38**	**1.46**	**7.89**	**0.42**	**17.11**	**1.07**	**18.20**	**1.23**	**19.28**
**Coeficient of Variation (%)**	**31.81**	**68.81**	**55.03**	**84.29**	**77.72**	**93.80**	**50.58**	**88.66**	**44.94**	**102.41**	**41.62**

**Table 2 plants-14-00392-t002:** Air pollutants removed annually by the vegetation of the studied cities. CO = carbon monoxide; NO_2_ = nitrogen dioxide; O_3_ = ozone; SO_2_ = sulfur dioxide; PM_2.5_ = particulate matter removed less than 2.5 microns; PM_10_ = particulate matter greater than 2.5 microns and less than 10 microns and t = tons.

City	CO(t)	NO_2_ (t)	O_3_ (t)	SO_2_ (t)	PM_2.5_ (t)	PM_10_ (t)
Aracaju (SE)	2.31	8.9	85.42	4.84	4.51	23.72
Belo Horizonte (MG)	7.46	28.8	276.27	15.65	14.6	76.71
Blumenau (SC)	13.32	51.39	492.95	27.92	26.05	136.87
Camaçari (BA)	9.21	35.53	340.8	19.3	18.01	94.62
Campos dos Goytacazes (RJ)	4.61	17.6	170.72	9.67	9.02	47.4
Colombo (PR)	2.71	10.47	100.44	5.69	5.31	27.89
Diadema (SP)	0.57	2.21	21.26	1.20	1.12	5.90
Duque de Caxias (RJ)	5.53	21.32	204.53	11.58	10.81	56.79
Feira de Santana (BA)	3.73	14.38	137.95	7.81	7.29	38.3
Ilhéus (BA)	1.22	4.69	45.03	2.55	2.38	12.5
Jaboatão dos Guararapes (PE)	2.93	11.29	108.32	6.14	5.72	30.07
Leme (SP)	0.41	1.59	15.33	0.86	0.81	4.25
Macaé (RJ)	1.87	7.21	69.14	3.92	3.65	19.2
Maceió (AL)	4.72	18.2	174.56	9.89	9.23	48.47
Mauá (SP)	1.41	5.42	52.03	2.95	2.75	14.45
Osasco (SP)	1.61	6.2	59.47	3.37	3.14	16.51
Porto Alegre (RS)	15.67	60.48	580.21	32.86	30.67	161.1
Recife (PE)	4.64	17.9	171.7	9.73	9.07	47.67
Santo André (SP)	1.88	7.27	69.75	3.95	3.69	19.37
São Bernardo do Campo (SP)	3.41	13.15	126.13	7.14	6.67	35.02
São Gonçalo (RJ)	4.6	17.74	170.23	9.64	9	47.26
Serra (ES)	4.34	16.75	160.71	9.1	8.49	44.62
Simões Filho (BA)	1.95	7.51	72	4.08	3.81	19.99
Suzano (SP)	1.68	6.48	62.13	3.52	3.28	17.25
Taboão da Serra (SP)	0.35	1.37	13.14	0.74	0.69	3.65
**Average**	**4.08**	**15.75**	**151.20**	**8.56**	**7.99**	**41.98**
**Coeficient of Variation (%)**	**93.33**	**93.39**	**93.34**	**93.35**	**93.36**	**93.35**

**Table 3 plants-14-00392-t003:** Carbon and CO_2_ Equivalent estimations for Brazilian cities studied. CO_2_ Eq = CO_2_ Equivalent.

Cities	Carbon (kt)	CO_2_ Eq (kt)	Carbon Stock (kt)	CO_2_ Eq Stock (kt)
Aracaju (SE)	5.32	19.5	109.85	402.78
Belo Horizonte (MG)	17.2	63.06	355.27	1302.64
Blumenau (SC)	30.69	112.52	633.91	2324.33
Camaçari (BA)	21.21	77.79	438.25	1606.9
Campos dos Goytacazes (RJ)	10.63	38.97	219.54	804.99
Colombo (PR)	6.25	22.92	129.15	473.56
Diadema (SP)	1.32	4.85	27.35	100.27
Duque de Caxias (RJ)	12.73	48.68	263.01	964.37
Feira de Santana (BA)	8.59	31.49	177.39	650.45
Ilhéus (BA)	2.8	10.28	57.91	212.32
Jaboatão dos Guararapes (PE)	6.74	24.72	139.29	510.72
Leme (SP)	0.95	3.5	19.72	72.31
Macaé (RJ)	4.3	15.78	88.91	326.01
Maceió (AL)	10.87	39.84	224.48	823.08
Mauá (SP)	3.24	11.88	66.91	245.33
Osasco (SP)	3.7	13.57	76.48	280.43
Porto Alegre (RS)	36.12	132.43	746.12	2735.78
Recife (PE)	10.69	39.19	220.8	809.58
Santo André (SP)	4.34	15.92	89.7	328.89
São Bernardo do Campo (SP)	7.85	28.79	162.2	594.73
São Gonçalo (RJ)	10.6	38.85	218.9	802.64
Serra (ES)	10	36.68	206.67	757.78
Simões Filho (BA)	4.48	16.43	92.59	339.51
Suzano (SP)	3.87	14.18	79.89	292.93
Taboão da Serra (SP)	0.81	3.01	16.90	61.99
**Average**	**9.41**	**34.59**	**194.44**	**712.97**
**Coeficient of Variation (%)**	**93.37**	**93.23**	**93.34**	**93.34**

**Table 4 plants-14-00392-t004:** Monetary estimation of the amount of CO_2_ Equivalent sequestered annually and the amount of CO_2_ Equivalent stocked in the vegetation of the cities studied. The values are in dollars.

City	Sequestred (USD)	Stocked (USD)
Aracaju (SE)	2,017,616.25	41,674,639.65
Belo Horizonte (MG)	6,524,660.55	134,780,904.20
Blumenau (SC)	11,642,163.10	240,492,614.30
Camaçari (BA)	8,048,736.83	166,261,925.80
Campos dos Goytacazes (RJ)	4,032,128.48	83,290,302.83
Colombo (PR)	2,371,475.1	48,998,069.30
Diadema (SP)	501,817.37	10,374,686.23
Duque de Caxias (RJ)	5,036,797.90	99,780,952.98
Feira de Santana (BA)	3,258,191.58	67,300,435.38
Ilhéus (BA)	1,063,645.90	21,968,219.60
Jaboatão dos Guararapes (PE)	2,557,716.60	52,842,921.60
Leme (SP)	362,136.25	7,481,734.92
Macaé (RJ)	1,632,717.15	33,731,439.68
Maceió (AL)	4,122,145.20	85,162,029.90
Mauá (SP)	1,229,193.90	25,383,681.78
Osasco (SP)	1,404,053.98	29,015,391.03
Porto Alegre (RS)	13,702,201	283,064,317.20
Recife (PE)	4,054,891.33	83,765,218.65
Santo André (SP)	1,647,202.60	34,029,426.08
São Bernardo do Campo (SP)	2,978,829.33	61,535,226.28
São Gonçalo (RJ)	4,019,712.38	83,047,154.20
Serra (ES)	3,795,187.90	78,405,602.15
Simões Filho (BA)	1,699,971.03	35,128,250.93
Suzano (SP)	1,467,169.15	30,308,734.78
Taboão da Serra (SP)	310,505.96	6,413,950.32
**Average**	**3,579,234.67**	**73,769,513.18**
**Coeficient of Variation (%)**	**93.22**	**93.63**

**Table 5 plants-14-00392-t005:** Population classes and cities that had their city center studied for the provision of ecosystem services.

Population	Cities
100,000–300,000	Leme (SP), Colombo (PR), Macaé (RJ), Simões Filho (BA) e Ilhéus (BA)
300,000–500,000	Suzano (SP), Diadema (SP), Taboão da Serra (SP), Blumenau (SC) e Camaçari (BA)
500,000–700,000	Campos dos Goytacazes (RJ), Mauá (SP), Aracaju (SE), Feira de Santana (BA) e Serra (ES)
700,000–900,000	São Bernardo do Campo (SP), Duque de Caxias (RJ), Osasco (SP), Jaboatão dos Guararapes (PE) e Santo André (SP)
>1 million	Recife (PE), São Gonçalo (RJ), Porto Alegre (RS), Belo Horizonte (MG), Maceió (AL)

**Table 6 plants-14-00392-t006:** Land use cover classes assessed for the provision of ecosystem services in 25 Brazilian cities.

Item	Cover Class	Description
IN	Impervious cover non-plantable no trees	sites where trees are impractical (road, rail, roof, monument)
IP	Impervious cover partially plantable no trees	Sites where trees are possible (sidewalk, parking lot, plaza)
SN	Soil cover non-plantable no trees	Sites where trees are impractical (dirt road)
SP	Soil cover partially plantable no trees	Soil cover partially plantable no trees
SVN	Short Vegetation cover non-plantable no trees	Grasses, herbaceous, or shrubs where trees are impractical (athletic field)
SVP	Short Vegetation cover partially plantable no trees	grasses, herbaceous, or shrubs where trees are possible (backyard)
TEI	Tree Evergreen over Impervious	trees with constant canopy over impervious ground
TEP	Tree Evergreen over Pervious	trees with constant canopy over pervious ground
W	Water	Ocean, estuary, river, lake, wetland, etc.
OTH	Other	areas not captured in above categories

## Data Availability

Data are contained within the article.

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
