# Peer review of "With Great Ecosystem Services Comes Great Responsibility: Benefits Provided by Urban Vegetation in Brazilian Cities"

_plants, 2025, doi:10.3390/plants14030392_

Round 1
Reviewer 1 Report
Comments and Suggestions for Authors
The manuscript deals with an interesting piece of work elated to ecosystems services. However, the authors have some scope to improve the manuscript, and here are my specific suggestions:
1. In the abstract, need to use the full meaning of IN (Page 1, line 23) in its first appearance.
2. The problem statement needs to be more specific and improved. I mean why this study is necessary and what was already discovered.
3. At the end of the introduction, the objectives were too many and better to make the objectives specific.
4. I don’t understand why the materials and methods section is placed after the discussion section. The materials and methods should be placed after the introduction section.
5. Ecosystem services are not only based on GIS or softer based data; it also need to include the reflection of local communities’ perceptions and their services and goods data as well. Also, there is no theoretical framework on which the ecosystem services/models rely on. Need to construct theoretical framework that would eventually guide the readers.
Good luck.
Author Response
Response to Reviewer 1 Comments
|
||
1. Summary |
|
|
We are writing to express our sincere gratitude for taking the time to review the manuscript. Your detailed and thoughtful feedback has been invaluable in improving the quality and clarity of our work. Your suggestions have provided us with important insights that have significantly enhanced the rigor and presentation of the research. We greatly appreciate your constructive criticism and the effort you dedicated to reviewing the paper. Once again, thank you for your time and valuable contribution.
|
||
2. Point-by-point response to Comments and Suggestions for Authors |
||
Comments 1: In the abstract, need to use the full meaning of IN (Page 1, line 23) in its first appearance. Response 1: Thank you for pointing that out. We have made the change.
Comments 2: The problem statement needs to be more specific and improved. I mean why this study is necessary and what was already discovered. Response 2: Thank you for your valuable feedback. To address your comment, we added lines 72-75. This addition specifies the threats posed by urbanization and references previous findings to contextualize the research.
Comments 3: At the end of the introduction, the objectives were too many and better to make the objectives specific. Response 3: Thank you for your helpful comment. The objectives have been revised to reduce their number and make them more specific (Lines 95-99). This revision condenses the objectives while maintaining their clarity and relevance to the study.
Comments 4: I don’t understand why the materials and methods section is placed after the discussion section. The materials and methods should be placed after the introduction section. Response 4: We have structured the paper in this way because it is the publication format adopted by the journal.
Comments 5: Ecosystem services are not only based on GIS or softer based data; it also need to include the reflection of local communities’ perceptions and their services and goods data as well. Also, there is no theoretical framework on which the ecosystem services/models rely on. Need to construct theoretical framework that would eventually guide the readers. Response 5: Although we recognize that the communities' perception of ecosystem services is important, it falls under cultural ecosystem services, which are not the focus of this study. In this article, we focus on regulating ecosystem services. |
Reviewer 2 Report
Comments and Suggestions for Authors
This paper quantifies the monetary benefits of urban vegetation using the value assessment method, thereby estimating the value of ecosystem services promoted by vegetation in 25 urban areas of the Brazilian Atlantic Forest (mainly including pollutant removal, carbon sequestration and carbon storage services). The i-Tree Canopy software was used to classify land use, and the Pearson algorithm was used to assess whether there is any relationship between socioeconomic variables and land cover types, pollutants and carbon sequestration. The full text has a clear viewpoint, a prominent center, and the discussion closely adheres to the theme. The selected evidence is sufficient and reasonable. The language expression is smooth, and the research method is appropriate. The topic of carbon fixation and carbon storage of urban vegetation is selected, closely adhering to the goals of carbon neutrality and enhancing human well-being proposed by the United Nations in recent years. The topic direction and problem selection are in line with the current global needs. At the same time, this study also points out the problems and challenges faced by urban green space area, which can inspire urban planners to consider innovative methods to create green spaces. Here are a few suggestions for this article:
1、 From the perspective of the article's first-level headings, "1. Introduction", "2. Results", "4. Materials and Methods", and "5. Conclusions", the third major part is missing. The "Discussion" section under "Results" can be proposed as a first-level heading and placed before "5. Conclusions". Within the "Discussion" section, it can be divided according to the hierarchy of the research results obtained, discussed in sequence, and secondary headings can be added to make the article's structure more distinct.
2、 The "4. Materials and Methods" section can be moved between the "1. Introduction" part and the "2. Results" part to serve as the second major section, making the article's logical structure more compact.
3、 The table formats in the text need to be consistent throughout. The formats of "Table 1, 2, 3, 4" can be adjusted to be the same as those of "Table 5, 6".
Comments on the Quality of English LanguageThe language expression is smooth, and the research method is appropriate.
Author Response
1. Summary |
|
|
We are writing to express our sincere gratitude for taking the time to review the manuscript. Your detailed and thoughtful feedback has been invaluable in improving the quality and clarity of our work. Your suggestions have provided us with important insights that have significantly enhanced the rigor and presentation of the research. We greatly appreciate your constructive criticism and the effort you dedicated to reviewing the paper. Once again, thank you for your time and valuable contribution.
|
||
2. Point-by-point response to Comments and Suggestions for Authors |
||
Comments 1: From the perspective of the article's first-level headings, "1. Introduction", "2. Results", "4. Materials and Methods", and "5. Conclusions", the third major part is missing. The "Discussion" section under "Results" can be proposed as a first-level heading and placed before "5. Conclusions". Within the "Discussion" section, it can be divided according to the hierarchy of the research results obtained, discussed in sequence, and secondary headings can be added to make the article's structure more distinct. Response 1: The authors would like to thank us for the missing first-level heading. We have made the correction. We have also made the adjustments by adding secondary headings to the “Discussions” section. The text is better organized. We appreciate this.
Comments 2: The "4. Materials and Methods" section can be moved between the "1. Introduction" part and the "2. Results" part to serve as the second major section, making the article's logical structure more compact. Response 2: We have structured the paper in this way because it is the publication format adopted by the journal.
Comments 3: The table formats in the text need to be consistent throughout. The formats of "Table 1, 2, 3, 4" can be adjusted to be the same as those of "Table 5, 6". Response 3: We have adjusted the format of the tables so that they are consistent throughout the text. We appreciate the review.
3. Response to comments on the quality of the English language Point 1: The English could be improved to express the research more clearly. |
Reviewer 3 Report
Comments and Suggestions for Authors
The paper in general lacks a clear purpose and context.
Prior to the results section there is only about a page and a half of introduction that contains a very limited amount of literature and theoretical framing of the work being discussed, why are you doing this study?
A discussion of the literature relating to the ecosystem services and urban forests is missing. The importance of them for carbon markets etc, and other socio-economic variables
Why do your methods come after your results and discussion - that doesn't make any sense? There also needs to be more detail on the methods. It's not really sufficient to simply, for example, "Since socioeconomic and population factors, such as wealth and population density, may correlate with the greater or lesser provision of green areas in cities (Chen et al., 2022; 469 Richards et al., 2017; Wu & Kim, 2021), we collected the following data from IBGE's "Cities" 470 portal (2023): total population, population density (Km²), Human Development Index, and 471 Gross Domestic Product per capita (R$)." - why these variables, for what purpose etc.
3. in line 478 you claim you use Pearson's test, but in the results you stated you used Spearman's - these are different tests. Also what was your hypothesis, were these data parametric (that would help explain which test was appropriate)?
4. The results are presented with a false amount of accuracy - there is no need for so many significant figures. Also the use of the data has some flaws - for example in line 306 you state that "the average 305 amount of CO2 sequestered annually among the cities studied is 59.89 tons, this may not be 306 representative of all cities" -this is problematic in many ways - i) I assume you mean tonnes, ii) if the cities are different sizes, in different climates and with different populations, this statistic is basically meaningless - do you mean per capita? per km2?
Author Response
Response to Reviewer 3 Comments
|
||
1. Summary |
|
|
We are writing to express our sincere gratitude for taking the time to review the manuscript. Your detailed and thoughtful feedback has been invaluable in improving the quality and clarity of our work. Your suggestions have provided us with important insights that have significantly enhanced the rigor and presentation of the research. We greatly appreciate your constructive criticism and the effort you dedicated to reviewing the paper. Once again, thank you for your time and valuable contribution.
|
||
2. Point-by-point response to Comments and Suggestions for Authors |
||
Comments 1: The paper in general lacks a clear purpose and context. Prior to the results section there is only about a page and a half of introduction that contains a very limited amount of literature and theoretical framing of the work being discussed, why are you doing this study? A discussion of the literature relating to the ecosystem services and urban forests is missing. The importance of them for carbon markets etc, and other socio-economic variables Response 1: Thank you for your feedback. To address this, the introduction was expanded to include a clearer statement of purpose, highlighting the study's significance in addressing global (Lines 42-52; 82-88; 96-99)
Comments 2: Why do your methods come after your results and discussion - that doesn't make any sense? Response 2: We have structured the paper in this way because it is the publication format adopted by the journal.
Comments 3: There also needs to be more detail on the methods. It's not really sufficient to simply, for example, "Since socioeconomic and population factors, such as wealth and population density, may correlate with the greater or lesser provision of green areas in cities (Chen et al., 2022; 469 Richards et al., 2017; Wu & Kim, 2021), we collected the following data from IBGE's "Cities" 470 portal (2023): total population, population density (Km²), Human Development Index, and 471 Gross Domestic Product per capita (R$)." - why these variables, for what purpose etc. Response 3: We have made adjustments and added more information to clarify this section (Lines 513-519; 521-533).
Comments 4: In line 478 you claim you use Pearson's test, but in the results you stated you used Spearman's - these are different tests. Also what was your hypothesis, were these data parametric (that would help explain which test was appropriate)? Response 4: The first version of the thesis used Pearson, but the analyses were changed to Spearman after the defence, as suggested by the participating professors. Unfortunately, we forgot to change the term throughout the text. Now we have adjusted the text.
Comments 5: The results are presented with a false amount of accuracy - there is no need for so many significant figures. Also the use of the data has some flaws - for example in line 306 you state that "the average 305 amount of CO2 sequestered annually among the cities studied is 59.89 tons, this may not be 306 representative of all cities" -this is problematic in many ways - i) I assume you mean tonnes, ii) if the cities are different sizes, in different climates and with different populations, this statistic is basically meaningless - do you mean per capita? per km2? Response 5: We deleted the sentence because, as the reviewer pointed out, it was meaningless. The authors decided to use tons instead of tonnes, since the two forms are variations between the imperial system and the metric system. If requested, we can modify this. |
Round 2
Reviewer 3 Report
Comments and Suggestions for Authors
Thanks for addressing the comments - the edits are appropriate to the issues I had. There is more detail and clarity in the methods - particularly the derivation of the land use classes.